# Environmental Impacts of Pastoral-Integrated Photovoltaic Power Plant in an Alpine Meadow on the Eastern Tibetan Plateau

Shaoying Wang[1, 2], Xianhong Meng[1, 2], Qian Li[3], Zhenchao Li[1,2], Peipei Yang[4], Wenzhen Niu[5], Lunyu Shang[1, 2]

[1]Key Laboratory of Cryospheric Science and Frozen Soil Engineering, Northwest Institute of Eco-Environment and Resources, Chinese Academy of Sciences, Lanzhou, 730000, China.

[2]Zoige Plateau Wetlands Ecosystem Research Station, Northwest Institute of Eco-Environment and Resources, Chinese Academy of Sciences, Lanzhou, 730000, China.

[3]Shannxi Province Climate Center, Xi'an, 710014, China.

[4]State Grid Lanzhou Electric Power Supply Company, Lanzhou, 730000, China.

[5]Wuling Power Ningxia Representative Office.

*Correspondence to*: Shaoying Wang (wangshaoying@lzb.ac.cn), Qian Li (liqian2011@163.com)

**Abstract:** Rising global energy demand and the transition to low-carbon sources have driven the rapid expansion of photovoltaic (PV) power plants, introducing significant land-use changes with largely unexplored ecological consequences. This study examined the microclimatic and soil hydrothermal impacts of a pastoral-integrated PV power plant in an alpine meadow ecosystem on the eastern Tibetan Plateau. Year-round observations from two 10-meter towers, located inside and outside the PV power plant, revealed that PV installations increased annual net radiation by 28.9%, while reducing albedo and wind speed by 31.6% and 36.2%, respectively. Air temperature responses were highly asymmetrical, with daytime and summer warming but nighttime and winter cooling. The PV power plant induced strong spatial heterogeneity in soil temperature and moisture: At 5cm depth, PV array gaps exhibited cold and moist conditions, whereas areas beneath the panels were cold and dry. These changes extended the soil frozen period by approximately 50 days and reduced the soil moisture depletion rate by 1.3 to 3.5 times compared to the reference site. These findings indicate that the PV power plant can alter local energy and water balances in ways that may buffer ecosystem responses to climate warming. However, further multi-year studies are needed to evaluate their long-term impacts on vegetation dynamics, carbon fluxes, and permafrost processes.

Keywords: Photovoltaic power plant, Alpine meadow, Microclimate, Soil hydrothermal dynamics, Field observations

# 1. Introduction

Photovoltaic (PV) power generation is a critical solution for addressing the global energy crisis, mitigating climate change, and reducing environmental pollution (Kan et al., 2021; Prăvălie et al., 2019). In recent decades, significant strides in the solar energy industry have been driven by the global transition from carbon-intensive fossil fuels to renewable energy and the rapid decline in solar PV costs (Wei et al., 2024). As the global leader in the photovoltaic industry, China has maintained its dominant position in PV power generation, with cumulative installed capacity accounting for approximately one-third of the global total (Birol, 2022). In line with its carbon peaking and carbon neutrality goals, China is expected to continue rapidly expanding PV power generation nationwide.

Despite the clear advantages of PV power plants in clean energy production, their widespread deployment significantly alters local land surface properties and climate (Armstrong et al., 2016; Wei et al., 2024). These changes result primarily from the combined effects of surface roughness, the dark surfaces of PV panels, their energy output, and heat released during power generation (Broadbent et al., 2019; Taha, 2013; Xu et al., 2024; Yang et al., 2017). However, the environmental impacts of PV power plants exhibit considerable regional variability, with studies reporting inconsistent trends and magnitudes of change depending on the local climate, ecosystem type, and PV power plants configuration.

For instance, PV power plants significantly influence albedo, a critical land surface parameter that directly influences the surface energy balance and climate dynamics (Wei et al., 2024). Numerical simulations often assume a simplified albedo value of 0.1 for PV power plants (Li et al., 2018; Lu et al., 2021), remote sensing and in-situ measurements typically report higher values, though with notable inconsistencies between the two methods (Chang et al., 2018; Li et al., 2022c; Wei et al., 2024; Xu et al., 2024; Yang et al., 2017). Similarly, PV power plants influence near-surface air temperatures, most field studies indicate that PV power plants increase daytime air temperatures due to heat released during electricity generation, a phenomenon similar

to the urban heat island effect (Armstrong et al., 2016; Broadbent et al., 2019; Fthenakis and Yu, 2013; Yang et al., 2017; Zheng et al., 2023). For example, Yang et al. (2017) showed that PV power plants in desert areas can increase both daytime and nighttime 2m air temperatures by approximately 0.7°C and 0.1°C, respectively, due to the heat released during power generation and the heat retention effect near the ground. In arid

regions of California, Barron-Gafford et al. (2016) found that PV power plants raised summer 2.5m air temperatures by more than 3°C compared to nearby wildlands. However, Keiko et al. (2009) conducted research on large-scale PV power plants in desert regions and found that PV modules had a self-cooling mechanism at night, with temperatures 2–4°C lower than the surrounding atmospheric temperature when sunlight

ceased. These significant variations in the environmental impacts of solar farms may be attributed to differences in their characteristics, such as type, spatial scale, capacity, installation methods, and background environmental conditions (Xu et al., 2024). These discrepancies highlight the need for observational studies to better understand the impacts of PV power plants across diverse climate zones and surface types.

The Tibetan Plateau (TP), known as the "Third Pole," is one of the most ecologically fragile regions on earth, playing a crucial role in global climate regulation. Its complex soil freeze-thaw dynamics, important water conservation functions, and substantial carbon release potential make it highly sensitive to environmental disturbances (Chen et al., 2016; Yao et al., 2022). Additionally, the TP's long daylight hours, high solar

radiation intensity, low temperatures and vast areas create ideal natural conditions for the development of PV industries (Li et al., 2022a; Tang et al., 2013; Wang and Qiu, 2009). In recent years, PV power plants have proliferated across the TP (Lyu et al., 2024), yet their impacts on microclimate and soil hydrothermal conditions, particularly in alpine meadows, remain underexplored. While extensive research exists on PV-

induced microclimatic changes in deserts, a critical gap remains in understanding their effects on alpine ecosystems, which are both ecologically fragile and climate-sensitive. To address this gap, this study investigates the effects of a pastoral-integrated PV power plant on the microclimate and soil hydrothermal conditions of an alpine meadow on the eastern TP. Field observations were conducted at two neighboring sites within the

Dongneng PV power plant to monitor key variables, including air temperature, humidity, radiation balance, and soil hydrothermal conditions. The primary aim of this research is to evaluate and quantify the environmental impacts of PV power plant deployment in this unique ecosystem, providing valuable insights into the environmental consequences of large-scale PV power plant. This work seeks to contribute to a deeper understanding of the environmental effects of renewable energy and offer insights for sustainable development as PV energy continues to expand.

## 2. Site and Method

### 2.1 Site description

The Dongneng PV power plant (34°3′19.5″N, 101°53′17.6″E) is situated in Maqu, Gansu Province, on the eastern Tibetan Plateau at an altitude of 3440 m (Figure 1a). The region's climate is classified as a sub-frigid humid zone based on China's climate regionalization (Zheng et al., 2010). The nearest meteorological station, located approximately 15 km from the study site, recorded a mean annual precipitation of 607 mm and an average air temperature of 1.84 °C from 1971 to 2020. The PV power plant, located on flat terrain, was constructed in September 2021 and became operational in August 2022. It has a capacity of 50 MW and utilizes bifacial photovoltaic panels (LONGi Green Energy Technology Co., Ltd.) with a photoelectric conversion efficiency of 21.1%. The arrays are south-facing, spaced 8 m apart, and fixed at an inclination angle of 36°. The panels are mounted 1.7 m and 4.4 m above the ground at their lower and upper edges, respectively.

This study deployed two 10-m towers (Fig. 01b): one within the PV power plant (PV site, Fig. 01c) and another reference site (RF site, Figure 1d) in an unmodified alpine meadow, approximately 180 m east of the PV power plant. Both sites were situated within the same contiguous open alpine meadow without physical barriers or fencing, and both are situated on flat terrain with similar slope and aspect. The average vegetation heights of 0.3 m in summer and 0.1 m in winter. Free-ranging sheep grazing by local herders was conducted across the entire area from May to September annually.

Given that both sites were subject to identical grazing regimes and stocking densities, potential impacts of trampling and grazing intensity on soil properties and vegetation were assumed to be comparable between the two locations.

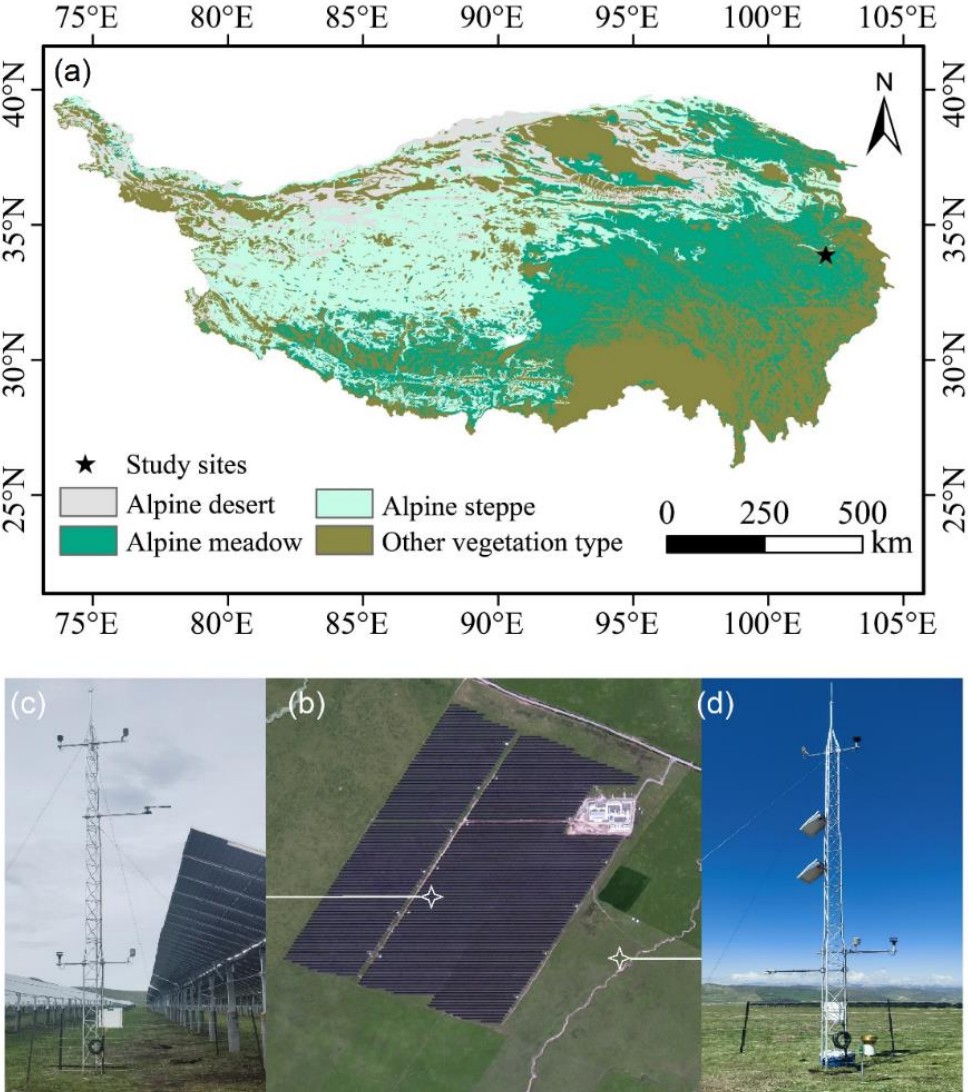

**Figure 1.** Map and photographs showing (a) location of the Dongneng solar power plant on the Tibetan Plateau. The study sites are marked with a star. (b) Google Earth image of the Dongneng solar power plant and its surrounding area. (c) Measurement tower installed within the PV power plant (PV site). (d) Measurement tower installed at the reference site (RF site) in an unmodified alpine meadow.

Soil samples collected from both the PV and RF sites were classified as sandy loam. At the RF site, bulk density was $0.71 \pm 0.07$ g cm⁻³ at 5 cm depth and $0.87 \pm 0.09$ g cm⁻³ at 10 cm depth. At the PV site, bulk density beneath the PV panels was $0.72 \pm 0.08$ g

cm⁻³ and 0.84±0.05 g cm⁻³ at 5 cm and 10 cm depths, respectively, while it was 0.76 ±0.10 g cm⁻³ and 0.99±0.05 g cm⁻³ for the area between the panel rows at the same depths. There were no significant differences in bulk density among positions at 5 cm depth ($p > 0.05$, paired t-test). However, at 10 cm depth, bulk density between the panel rows was significantly higher than that beneath the PV panels and at the RF site ($p < 0.05$, paired t-test). The slightly higher bulk density at the PV site may result from anthropogenic soil compaction associated with construction and maintenance activities. Vegetation surveys conducted during the peak growing season using 0.3 m × 0.3 m quadrats showed vegetation cover exceeded 95% at all positions, with *Stipa aliena*, *Potentilla anserina*, and *Scirpus pumilus* identified as the dominant species.

## 2.2 Measurements

The RF site served as a baseline for comparing environmental conditions in areas impacted by the PV power plant. Air temperature and humidity were recorded using HC2A-S3 sensors (Rotronic Instrument Corp., Switzerland) at 2.5 m and 10 m heights on both towers. Wind speed and direction were measured with WindSonic4 sensors (Gill Instruments, UK) at the same heights to evaluate the PV power plant's effect on local airflow patterns. Four-component radiation measurements (CNR-4, Kipp and Zonen, Netherlands) were taken at 2 m for the RF site and at 7 m for the PV site to assess the differences in radiation balance above the PV power plant and over the natural meadow. Soil temperature (CS109, Campbell Scientific, Inc., USA) and moisture (CS616, Campbell Scientific, Inc., USA) were measured at depths of 5 cm and 10 cm. At the PV site, sensors were installed beneath the PV panels and in the inter-row gaps to assess the hydrothermal effects of shading. All sensors were set to a sampling frequency of 1 Hz, with data averaged every 10 minutes by the CR1000 data loggers (Campbell Scientific, Inc., USA). Data were collected continuously over a one-year period, from June 2023 to May 2024, to capture seasonal variations in microclimate and soil hydrothermal conditions.

## 2.3 Data processing

To ensure data quality, the study applied the following quality control measures: (1) short-wave radiation values at night were set to zero according to the solar altitude angle;

(2) daytime downward shortwave radiation values exceeding the solar constant (1361 W/m² ) were removed; and (3) the median absolute deviation (MAD) method (Mauder et al., 2013) were used to detect outlier in temperature and humidity data. If three or more consecutive outliers were detected, they were not considered as actual anomalies. As a result of these procedures, over 99% of the 10-minute microclimatic and soil hydrothermal data from June 2023 to May 2024 were retained, with no continuous gaps exceeding 1 hours. Missing values were filled via linear interpolation using a centered 12-point (2-hour) moving window. Daily means were calculated only when at least 75% of the 10-minute data were available. Seasonal and annual means were then computed from these daily averages without further interpolation.

## 2.4 Method

Net radiation (Rn) was calculated as the sum of downward shortwave radiation (DR) and downward longwave radiation (DLR), minus upward shortwave radiation (UR) and upward longwave radiation (ULR):

$$Rn = DR + DLR - UR - ULR \qquad (1)$$

Corresponding, surface albedo ($\alpha$) was calculated as the ratio of upward to downward shortwave radiation:

$$\alpha = UR / DR \qquad (2)$$

To assess the statistical significance of differences between the PV and RF sites, paired Student's t-tests were conducted for seasonal and annual averages of key microclimatic and soil variables.

To assess the relative importance of environmental drivers influencing soil moisture dynamics during dry periods, we applied a Boosted Regression Tree (BRT) modeling approach (Elith et al., 2008). BRT is an ensemble machine learning method that combines the strengths of regression trees and boosting algorithms, allowing for flexible modeling of nonlinear relationships and high-order interactions among

predictors. The response variable was the daily mean soil water content, while predictor variables included near-surface air temperature, vapor pressure deficit, wind speed, and shallow soil temperature. The analysis was performed separately for the RF site, the PV array gap, and beneath-panel locations. All predictors were standardized prior to model fitting, and model hyperparameters were optimized to minimize predictive deviance using 10-fold cross-validation.

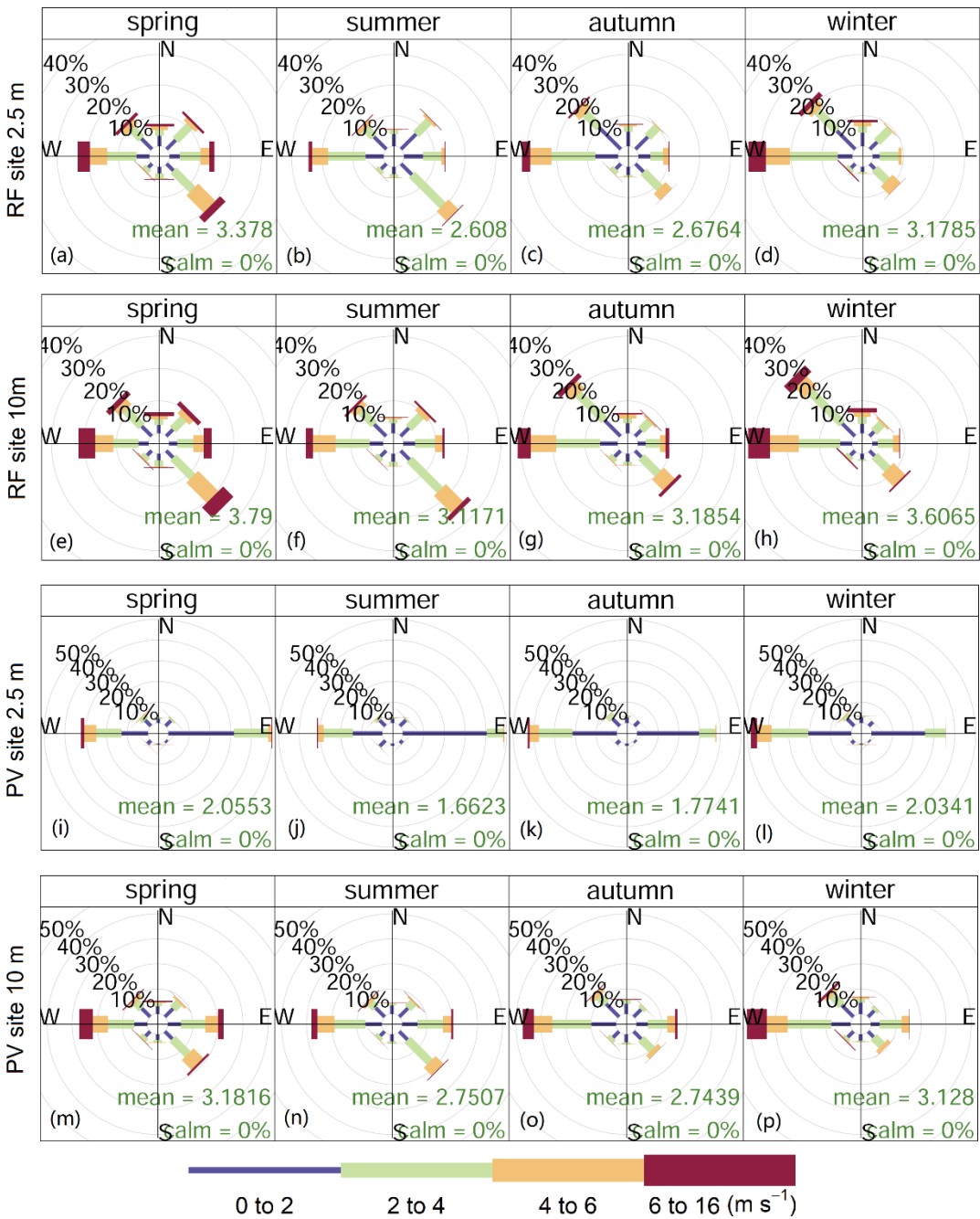

**Figure 2.** Seasonal wind roses at 2.5 m and 10 m heights for the RF site (a–h) and PV site (i–p) during spring, summer, autumn, and winter. The mean wind speed (m/s) is shown below each wind rose.

## 3. Results and discussion

### 3.1 The effect of PV power plant on wind regimes

During the observation period, the PV site experienced prevailing winds predominantly from the west and southeast (Figure 2). The structured layout of the PV panels redirected airflow primarily toward the east and west, contrasting with the more variable wind directions observed at the RF site. At a height of 2.5 m, beneath the upper edge of the PV panels, west and east winds accounted for 48% of total airflow in summer and 57% in winter, compared to 38% and 43% at the RF site.

The presence of PV panels increased surface roughness, enhancing frictional drag and obstructing near-surface wind flow. Consequently, wind speeds at a height of 2.5 m decreased by 39.1% in spring, 36.0% in summer, 33.6% in fall, and 36.2% in winter ($p < 0.001$ for all seasons, paired t-test). The reduction was most pronounced for southeast winds, exceeding 70% ($p < 0.001$, paired t-test), while the impact on west winds was relatively minor, with wind speed reductions of approximately 30% ($p < 0.001$, paired t-test).

At 10 m, above the top edge of the PV panels, the directional redistribution of airflow by the panels remained consistent. West and east winds accounted for 48% of the total in summer and 56% in winter, compared to 36% and 40% at the RF site. However, the obstructive effect of the PV panels on wind speed was also significant at this height, with reductions of 15.8% in spring, 11.9% in summer, 14.1% in fall, and 13.3% in winter. ($p < 0.01$ for all seasons, paired t-test) North winds experienced the largest reductions (approximately 40%) ($p < 0.001$, paired t-test), while west winds were minimally affected, with reductions of around 7% ($p < 0.05$, paired t-test).

These findings reveal a pronounced directional dependence of the PV panels' influence on wind regimes, consistent with previous studies (Jiang et al., 2021; Li et al., 2022b). However, variations in the magnitude of effects can be attributed to differences in PV

field layouts and background climatic conditions.

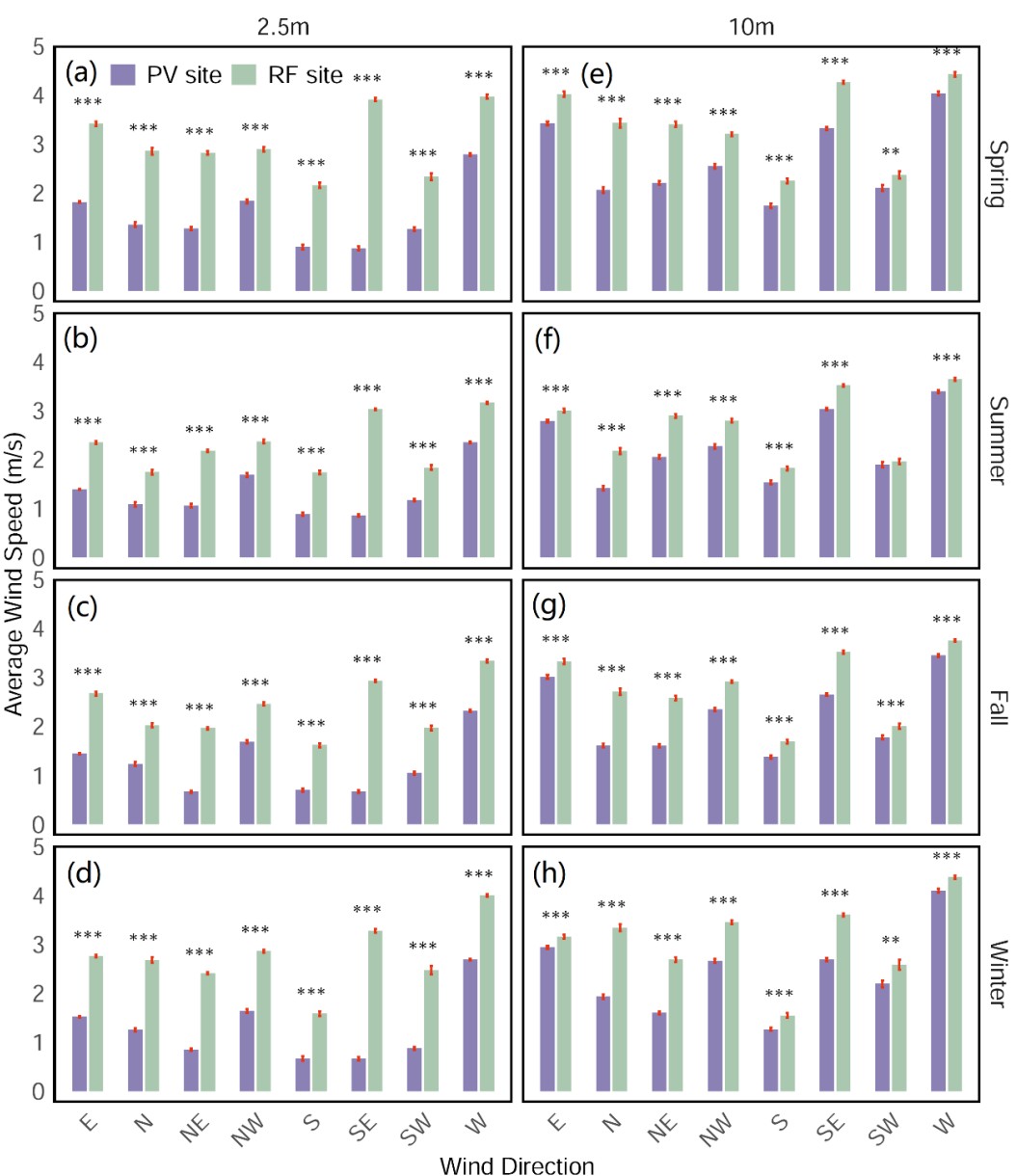

**Figure 3.** Seasonal variations in average wind speed (m/s) by direction at 2.5 m (a–d) and 10 m (e–h) heights for the PV site and RF site during spring, summer, autumn, and winter. Error bars represent standard deviations, and asterisks indicate statistically significant differences (* $p <$ 0.05, ** $p <$ 0.01, *** $p <$ 0.001).

## 3.2. The effect of PV power plant on surface radiation components

As shown in Figure 4 and Table 1, downward shortwave radiation (DR) and downward longwave radiation (DLR) measured above the PV panels were comparable to those observed over the natural meadow, with no statistically significant differences in daily totals across all seasons ($p > 0.05$, paired t-test). However, upward shortwave radiation

(UR) was substantially reduced at the PV site due to the strong solar absorption of the PV panels. This reduction was particularly pronounced in the peak values of the

260 seasonal average diurnal variations. The peak UR values were approximately 38% lower in summer and 50% lower in winter at the PV site (p<0.001, paired t-test). The daily total UR at the PV site was significantly lower than at the RF site across all seasons (p<0.001, paired t-test), with reductions of 36.7% in spring, 36.8% in summer, 43.2% in fall, and 47.8% in winter. On an annual scale, the UR at the PV site was reduced by

265 40.8% compared to the RF site (p < 0.001, paired t-test).

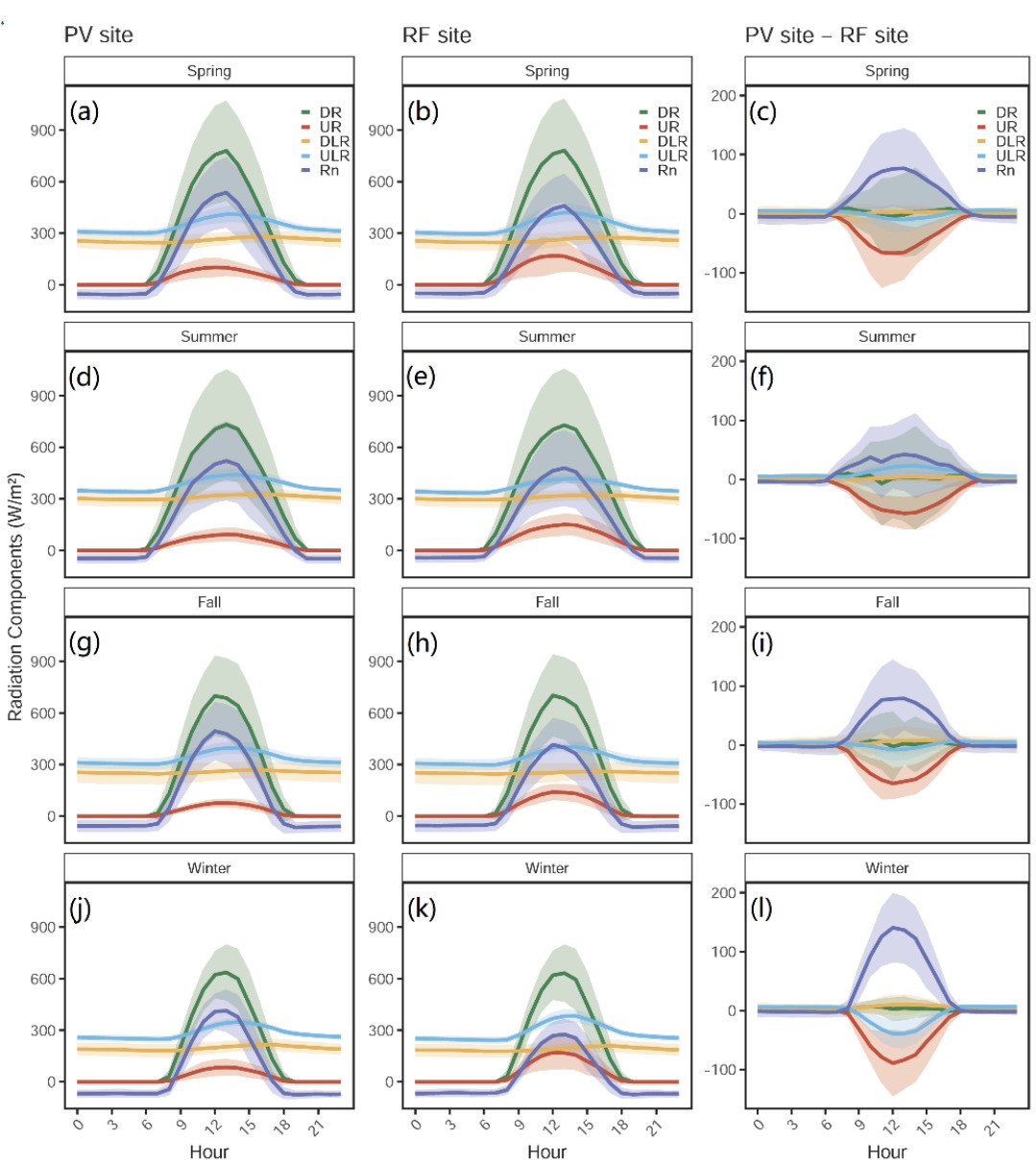

**Figure 4.** Seasonal averaged diurnal variations in radiation components at the PV site (a, d, g, j), RF site (b, e, h, k), and their differences (c, f, i, l) during spring, summer, autumn, and winter.

Radiation components include downward shortwave radiation (DR), upward shortwave radiation (UR), downward longwave radiation (DLR), upward longwave radiation (ULR), and net radiation (Rn). Shaded areas around the lines represent standard deviations.

**Table 1.** Averaged daily total radiation (MJ) components for PV and RF sites. Statistical significance between sites for each radiation component is denoted by asterisks (* $p < 0.05$, ** $p < 0.01$, *** $p < 0.001$)

| | | PV site | RF site | Relative change |
|---|---|---|---|---|
| DR | Spring | 20.48±5.79 | 20.33±5.92 | 0.7% |
| | Summer | 20.99±7.42 | 20.82±7.50 | 0.8% |
| | Fall | 16.94±5.08 | 16.83±5.13 | 0.6% |
| | Winter | 14.12±3.13 | 14.00±3.21 | 0.9% |
| | Annual | 18.13±6.20 | 17.99±6.26 | 0.8% |
| UR | Spring | 2.93±1.25 | 4.63±2.01 | -36.7%*** |
| | Summer | 2.85±1.05 | 4.51±1.63 | -36.8%*** |
| | Fall | 2.04±0.67 | 3.59±1.13 | -43.2%*** |
| | Winter | 2.04±1.18 | 3.90±2.05 | -47.8%*** |
| | Annual | 2.46±1.14 | 4.16±1.79 | -40.8%*** |
| DLR | Spring | 22.54±2.76 | 22.46±3.03 | 0.4% |
| | Summer | 26.77±3.65 | 26.59±3.73 | 0.7% |
| | Fall | 22.15±4.49 | 21.95±4.70 | 0.9% |
| | Winter | 15.94±2.42 | 15.81±2.42 | 0.8% |
| | Annual | 21.85±4.80 | 21.70±5.00 | 0.7% |
| ULR | Spring | 29.52±1.92 | 29.41±1.77 | 0.4 %** |
| | Summer | 32.12±3.54 | 31.90±3.40 | 0.7%** |
| | Fall | 29.04±2.54 | 28.88±2.21 | 0.6%** |
| | Winter | 24.48±1.04 | 24.85±1.03 | -1.5%*** |
| | Annual | 28.79±3.76 | 28.76±3.22 | 0.1%** |
| Rn | Spring | 10.56±3.15 | 8.75±3.06 | 20.7%*** |
| | Summer | 12.79±4.18 | 11.00±3.77 | 16.3%*** |
| | Fall | 8.01±3.58 | 6.31±3.69 | 26.8%*** |
| | Winter | 3.55±2.11 | 1.05±1.42 | 238.1%*** |
| | Annual | 8.73±4.39 | 6.78±4.63 | 28.8%*** |

The daily surface albedo (α) at the PV site was 0.159±0.074 in spring, 0.144±0.013 in summer, 0.139±0.027 in fall, and 0.168±0.073 in winter. Compared to the RF site, the PV power plant significantly reduced the albedo by 34.4%, 36.9%, 38.5%, and 41.5%, respectively ($p < 0.001$ for all seasons, paired t-test). At an annual scale, installation of a PV power plant led to a 31.6% decrease in surface albedo compared to the alpine meadow ($p < 0.001$, paired t-test). As summarized in Table 2, the reduction in albedo of this alpine meadow is slightly lower than the average findings over desert areas at

annual and seasonal scales (Broadbent et al., 2019; Li et al., 2022c; Stern et al., 2023; Yang et al., 2017; Ying et al., 2023), but much higher than that observed over barren areas (Chang et al., 2018) and water body (Ying et al., 2023). The reduction in this study aligns with the finding that the higher the albedo of the background surface, the more pronounced the relative change in surface albedo over the PV power plant (Xu et al., 2024).

For upward longwave radiation (ULR), although the daily cumulative values between the two sites are relatively similar (Table 1), the differences in seasonal and annual averages are statistically significant ($p < 0.01$, paired t-test). Additionally, their variations differ between daytime and nighttime. During nighttime, ULR at the PV site is consistently slightly higher than that at the RF site across all seasons, with an average increase of approximately 5.0 $Wm^{-2}$. However, during the daytime, seasonal variations showed contrasting patterns. In spring, fall, and winter, the PV site recorded lower ULR than the RF site. The daytime greatest negative deviation being most pronounced in winter (-39.0 $W\ m^{-2}$), followed by spring (-10.1 $Wm^{-2}$) and autumn (-8.0 $Wm^{-2}$).

PV power plant effects on ULR, which vary between daytime and nighttime as well as across different seasons, have also been observed in previous studies (Broadbent et al., 2019; Chang et al., 2018; Jiang et al., 2021; Li et al., 2022b; Yang et al., 2017). The underlying reasons for these variations are primarily attributed to three interrelated factors: (1) the lower emissivity of PV modules (~0.83) (Broadbent et al., 2019) compared to natural surface (0.95–1) (Campbell and Norman, 1998), which reduce ULR; 2) the cavity effect (Broadbent et al., 2019), where semi-enclosed spaces beneath PV modules cause repeated radiative exchanges, enhancing ULR; and 3) differences in land surface temperature (LST), which influences ULR patterns depending on the season and time of day(Chang et al., 2018; Yang et al., 2017). Wang et al. (2024b) reported that the PV site exhibited higher land surface temperatures (LST) during nighttime and the warm-season daytime, but lower LST during the daytime in the cold season. This result suggests that during both nighttime and warm-season daytime, the elevated LST at the PV site exerts a positive influence on ULR, while the reduced LST at the PV site during the cold-season daytime has a negative effect, diminishing ULR.

In this study, we infer that the higher ULR observed at the PV site during both nighttime and summer daytime is primarily due to the combined positive effects of the cavity effect and the higher LST, which outweigh the negative effect of the PV modules' lower emissivity. Conversely, the lower ULR at the PV site compared to the reference site during the daytime in spring, autumn, and winter likely results from the dominant negative effects of lower LST and the PV modules' lower emissivity, which surpass the positive effect of the cavity effect.

**Table 2.** Comparison of in-situ albedo observations across different PV power plant sites.

| Location | Land cover | In-situ albedo observations | | | Source |
|---|---|---|---|---|---|
| | | Background | Absolute change | Relative change | |
| 36.136°N 100.588°E | Barren | 0.17$^{Summer}$ | -0.01 | -6.30% | Chang et al. (2018) |
| 36.136°N 100.588°E | Barren | 0.19$^{Winter}$ | 0.02 | 11% | Chang et al. (2018) |
| 36.503°N 95.233°E | Desert | 0.26$^{Annual}$ | -0.07 | -27% | Yang et al. (2017) |
| 44.410°N 87.660°E | Desert | 0.23$^{Summer}$ | -0.09 | -39.10% | Li et al. (2022c) |
| 44.410°N 87.660°E | Desert | 0.22$^{Summer}$ | -0.08 | -36.40% | Ying et al. (2023) |
| 29.965°N 35.059°E | Desert | 0.38$^{Annual}$ | -0.21 | -55.20% | Stern et al. (2023) |
| 32.555°N 111.284°W | Desert | 0.31$^{October\ to\ June}$ | -0.11 | -35.50% | Broadbent et al. (2019) |
| 32.303°N 119.793°E | Water body | 0.101$^{annual}$ | -0.019 | -18.80% | Li et al. (2022b) |
| 34.055°N 101.888°E | Alpine meadow | 0.242±0.093$^{Spring}$ | -0.083 | -34.40% | This study |
| 34.055°N 101.888°E | Alpine meadow | 0.229±0.017$^{Summer}$ | -0.084 | -36.90% | This study |
| 34.055°N 101.888°E | Alpine meadow | 0.227±0.032$^{Fall}$ | -0.087 | -38.50% | This study |
| 34.055°N 101.888°E | Alpine meadow | 0.288±0.101$^{Winter}$ | -0.12 | -41.50% | This study |
| 34.055°N 101.888°E | Alpine meadow | 0.246±0.075$^{annual}$ | -0.085 | -40.00% | This study |

The net radiation (Rn) differences between the PV and RF sites also showed clear

seasonal dependencies. The PV site exhibited higher Rn across all seasons (p < 0.001 paired t-test), with peak diurnal variations exceeding the RF site by 76.8 Wm$^{-2}$ in spring, 42.0 Wm$^{-2}$ in summer, 78.8 Wm$^{-2}$ in fall, and 140.6 Wm$^{-2}$ in winter. The relative difference in Rn between two sites also showed that the influence of PV power plant on Rn was most pronounced in winter (Table 1), corresponds to the result that the largest relative difference in albedo during this season. Consistent with previous studies (Broadbent et al., 2019; Chang et al., 2018; Jiang et al., 2021; Li et al., 2022b; Li et al., 2022c), PV modules can enhance land surface energy availability. The relative difference in Rn between our two sites showed that the influence of PV power plant on Rn was most pronounced in winter (Table 1), not only due to the largest relative difference in albedo during this season but also as a result of manual adjustments to the tilt angle of PV panels and snow clearing on their surfaces.

**3.3 The effect of PV power plant on air temperature and humidity**

The PV power plant significantly influenced air temperature (Ta) at both 2.5 m and 10 m heights, with diurnal and seasonal variations observed between the PV site and RF site (Figure 5). At 2.5 m, the PV site exhibited an average annual Ta increase of 0.08°C compared to the RF site (p < 0.001, paired t-test). This increase was more pronounced at 10 m, with an annual mean difference of 0.17°C (p < 0.001, paired t-test).

The influence of PV power plant on Ta demonstrated a distinct diurnal asymmetry. During the daytime, the PV site consistently shows a warm bias relative to the RF site due to heat released from PV panels. The maximum warm bias ranges from 0.65°C (winter) to 1.60°C (summer) at 2.5 m, and from 0.46°C (winter) to 0.70°C (summer) at 10 m. These values are consistent with previous studies in desert or barren areas, which reported local increases in daytime air temperature ranging from 0.1°C to 1.9°C (Broadbent et al., 2019; Fthenakis and Yu, 2013; Jiang et al., 2021; Yang et al., 2017). During the nighttime, the PV site exhibits a cold bias relative to the RF site. At 2.5 m, the maximum cold bias ranges from -0.46°C in summer to -0.89°C in winter, while at 10 m, it varies between -0.01°C in summer and -0.46°C in winter. The cooling effect of PV power plant is primarily attributed to two key mechanisms (Barron-Gafford et al.,

2016; Broadbent et al., 2019; Yang et al., 2017): (1) the shielding effect of PV panels, which minimizes heat accumulation in the soil and enhances nocturnal radiative cooling of the ground surface; and (2) the reduction in PV panel temperatures below ambient air temperature at night, further contributing to the cooling of near-surface air. Interestingly, a small number of studies have reported that PV power plants also exhibit a nocturnal warming effect. For instance, Barron-Gafford et al. (2016) observed a significant nocturnal heating effect of 3–4°C at a height of 2.5 m in a utility-scale PV power plant in southern Arizona. This may be attributed to the presence of impervious surfaces near the PV power plant, as well as the smaller and less continuous scale of the PV power plant (Broadbent et al., 2019).

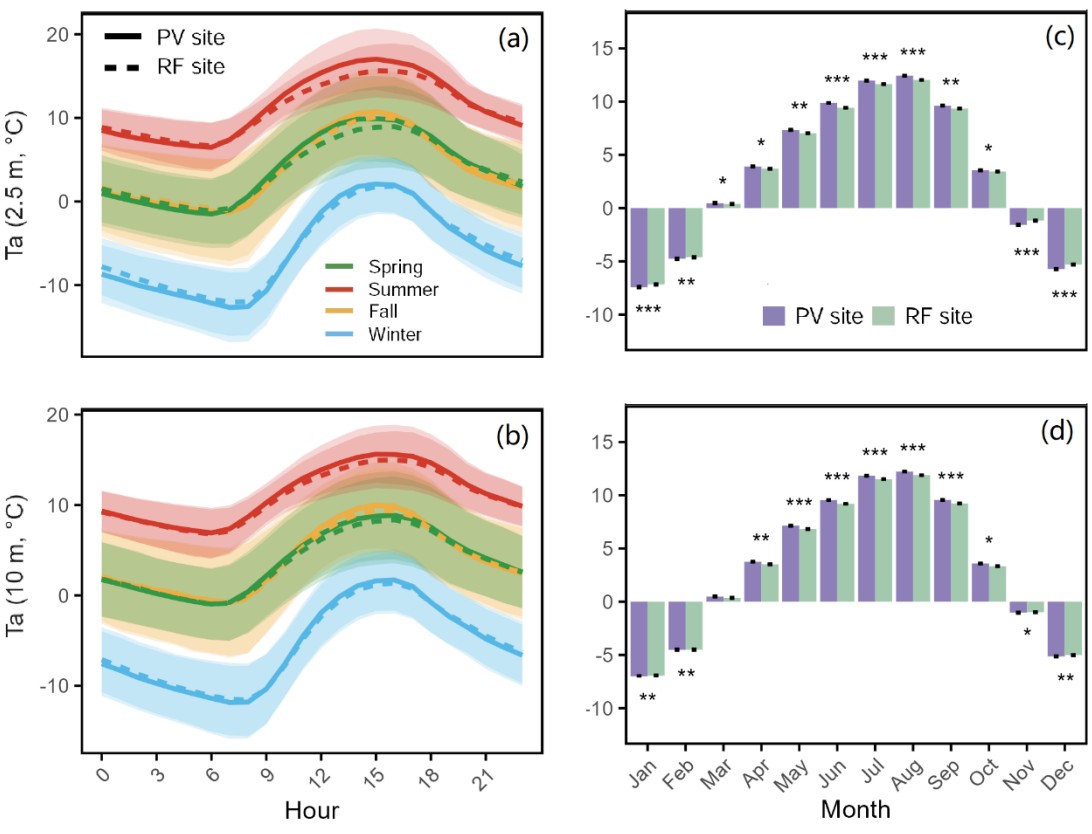

**Figure 5.** Diurnal (a, b) and monthly variations (c, d) in air temperature (Ta) at 2.5 m and 10 m height for PV and RF sites. The shaded areas around the lines represent standard deviation, and asterisks indicate statistically significant differences (* $p < 0.05$, ** $p < 0.01$, *** $p < 0.001$).

The dynamics of monthly Ta suggest that PV power plant also exhibit asymmetrical effects on near-surface air temperature between warm and cold seasons (Figure 5c, d). From March to October, the PV site generally experiences significantly higher Ta than

the RF site (p < 0.001, paired t-test). However, from November to February, the PV site sees lower Ta compared to the RF site (p < 0.01, paired t-test). This seasonal reversal may be attributed to the enhanced cooling effect and reduced warming effect of PV

panels during the colder months with lower solar input.

The warming effect of PV power plant on Ta in this alpine meadow region is much weaker than that observed in low-altitude areas (Barron-Gafford et al., 2016; Fthenakis and Yu, 2013; Jiang et al., 2021; Li et al., 2022b; Yang et al., 2017). This phenomenon can be attributed not only to differences in PV power plant characteristics but, more

importantly, to the lower background temperature in this region. Higher daily temperatures are known to enhance the warming effect of PV power plant (Broadbent et al., 2019; Jiang et al., 2021).

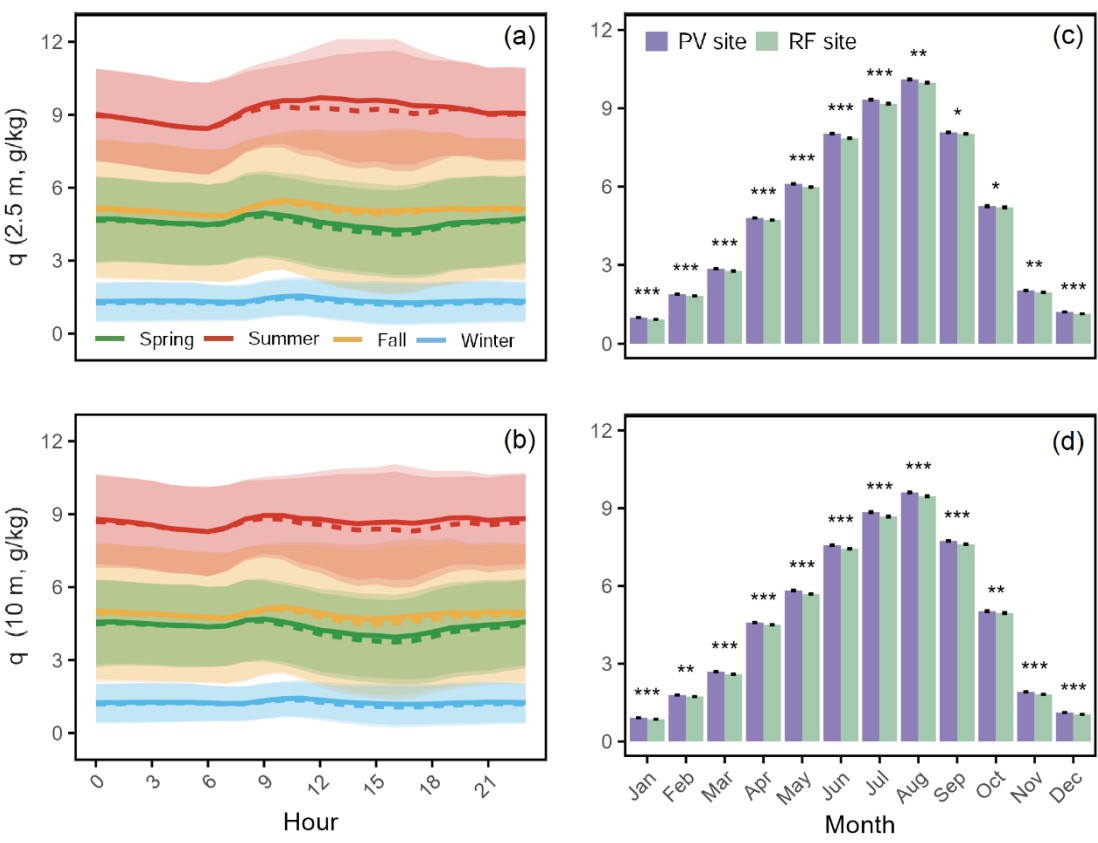

**Figure 6.** Diurnal (a, b) and monthly variations (c, d) in air specific humidity (q) at 2.5 m and 10
380 m height for PV and RF sites. The shaded areas around the lines represent standard deviation, and asterisks indicate statistically significant differences (* p < 0.05, ** p < 0.01, *** p < 0.001).

Specific humidity (q) was also influenced by the PV power plant. It was consistently higher than at the RF site (Figure 6). This increase was most significant during summer

daytime (p < 0.001, paired t-test), when q at the PV site was up to 4% higher at a height

of 2.5 m. This can be attributed to reduced wind speeds and lower evapotranspiration

beneath the PV panels, which retain more moisture in the immediate environment. At

10 m, however, the differences in q between the two sites were less pronounced due to

the reduced influence of near-surface shading.

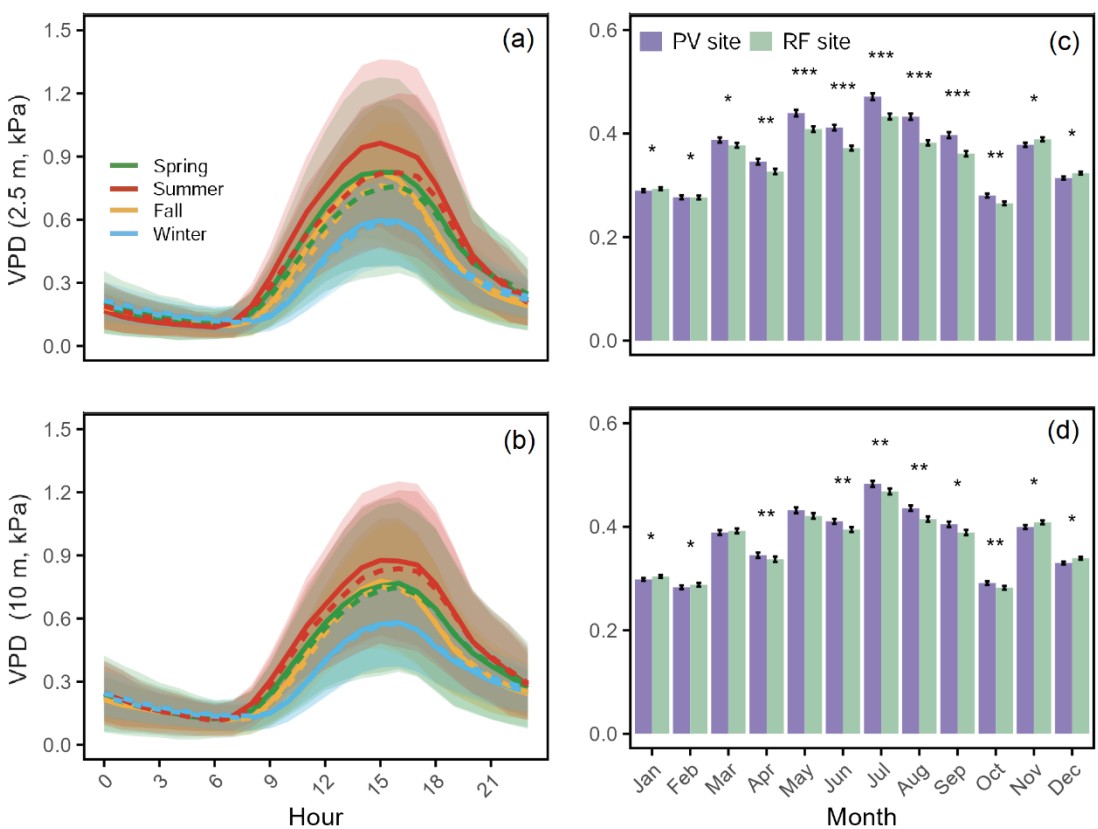

**Figure 7.** Diurnal (a, b) and monthly variations (c, d) in vapor pressure deficit (VPD) at 2.5 m and
10 m height for PV and RF sites. The shaded areas around the lines represent standard deviation,
and asterisks indicate statistically significant differences (* p < 0.05, ** p < 0.01, *** p < 0.001).

The vapor pressure deficit (VPD) exhibits diurnal and seasonal patterns (Figure 7)

similar to those of Ta (Figure 5). The daytime maximum positive bias ranges from 0.03

395    kPa (winter) to 0.16 kPa (summer) at 2.5 m, and from 0.01 kPa (winter) to 0.05 kPa

(summer) at 10 m. The nighttime maximum negative bias occurs in spring, with a value

less than 0.03 kPa. On a seasonal scale, the difference in VPD between the two sites

reaches its peak in summer, with the average VPD at 2.5 m and 10 m at the PV site

being approximately 10.8% (p < 0.001, paired t-test) and 4.1% (p < 0.01, paired t-test)

higher than at the RF site, respectively. Annually, VPD at the PV site is slightly higher

than at the RF site, with differences of about 5% at 2.5 m (p < 0.001, paired t-test) and 1% at 10 m (p < 0.001, paired t-test).

The discrepancy between VPD and q suggests that variations in VPD are primarily driven by temperature changes rather than differences in specific humidity in this alpine region. This indicates that, despite the higher q at the PV site, the warming effect of PV power plant during the daytime elevates the air's evaporative demand, potentially exacerbating water loss from vegetation and soil. However, the relatively higher soil moisture observed at the PV site may help offset this effect by sustaining local evaporation and transpiration rates.

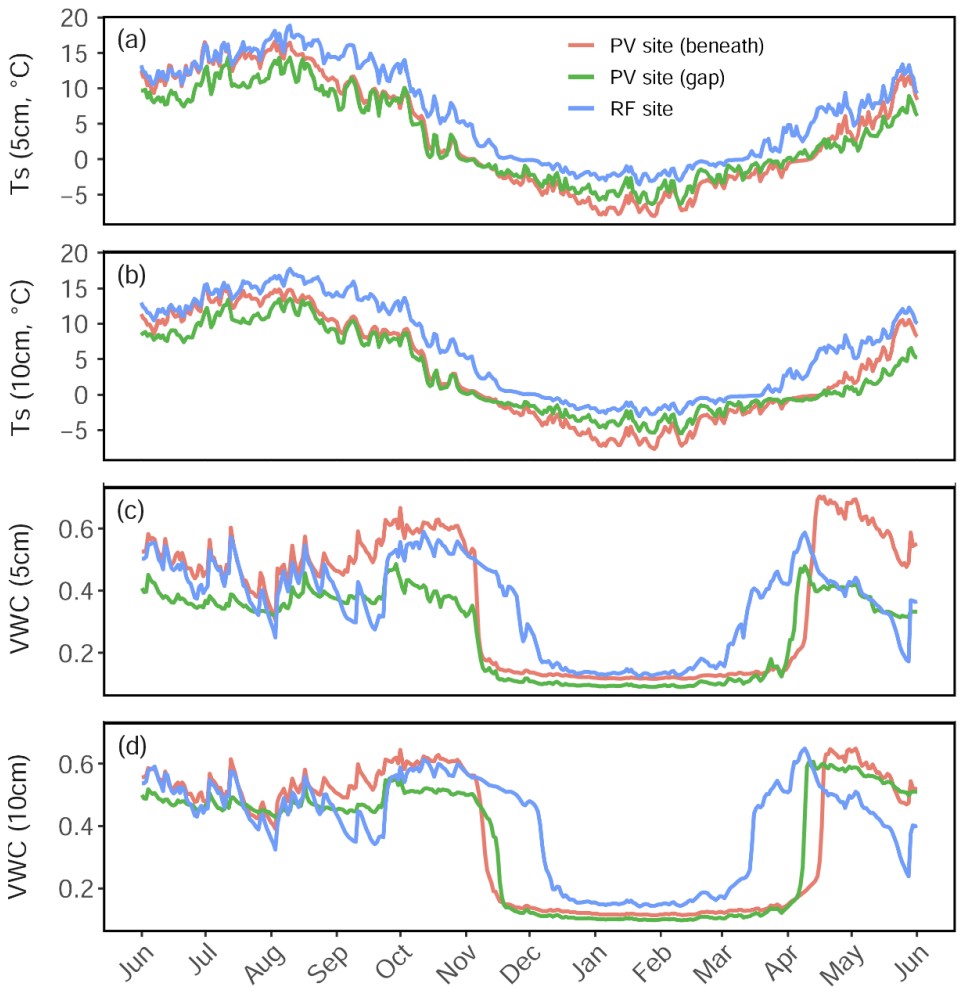

**Figure 8.** Seasonal variations in (a, b) soil temperature (Ts) and (c, d) volumetric water content (VWC) at 5 cm and 10 cm depths for the PV site (beneath panels and in gaps) and the RF site.

### 3.4. The effect of PV power plant on soil temperature and soil moisture

The presence of PV power plant significantly influences the thermal properties of the

soil. The shading effect of PV power plant results in consistently lower soil temperatures (Ts) beneath the panels and in the gaps between rows throughout the year. Soil beneath the panels begin to freeze (daily Ts below 0°C for three consecutive days) 27 days earlier and thaw 20 days later compared to the RF site. In the gaps between rows, freezing begins 20 days earlier and thawing is delayed by 31 days. At a 10 cm depth, similar trends are observed, with freezing starting 28 days earlier and thawing delayed by 31 days beneath the panels, and in the gaps, freezing occurs 26 days earlier with a 28-day delay in thawing.

**Table 3.** Average soil temperature (°C) at 5 cm and 10 cm depths for PV and RF sites.

|  | Annual | Spring | Summer | Fall | Winter |
|---|---|---|---|---|---|
| RF site | 6.50 | 5.29 | 14.64 | 7.73 | -1.67 |
| (5cm) | (±0.031) | (±0.040) | (±0.025) | (±0.048) | (±0.011) |
| PV site | 3.53 | 2.24 | 13.50 | 3.89 | -5.50 |
| (gap, 5cm) | (±0.034) | (±0.041) | (±0.024) | (±0.043) | (±0.015) |
| PV site | 2.97 | 1.65 | 10.73 | 3.50 | -4.00 |
| (beneath, 5cm) | (±0.027) | (±0.028) | (±0.024) | (±0.040) | (±0.015) |
| RF site | 6.37 | 4.66 | 14.37 | 7.95 | -1.49 |
| (10cm) | (±0.029) | (±0.035) | (±0.019) | (±0.045) | (±0.008) |
| PV site | 3.27 | 1.53 | 12.61 | 4.08 | -5.14 |
| (gap, 10cm) | (±0.031) | (±0.035) | (±0.017) | (±0.040) | (±0.014) |
| PV site | 2.79 | 0.48 | 10.42 | 3.59 | -3.34 |
| (beneath, 10cm) | (±0.025) | (±0.020) | (±0.017) | (±0.036) | (±0.011) |

During spring, summer, and fall, Ts at both depths are highest at the RF site, followed by the gaps between the rows, and lowest beneath the panels (Table 3). The differences in Ts between the locations are most pronounced in the fall, with the RF site showing that Ts is 3.89°C higher than in the gaps ($p < 0.001$, paired t-test) and 4.23°C higher than beneath the panels ($p < 0.001$, paired t-test) at a depth of 5 cm. This result aligns with previous studies that have documented significant cooling effects in PV power plant due to shading, which lowers the soil temperature relative to surrounding areas (Armstrong et al., 2016; Choi et al., 2024; Wu et al., 2022; Yue et al., 2021; Zheng et al., 2023).

Wintertime Ts shows a distinct spatial pattern, with values being highest at the RF site, intermediate beneath the panels, and lowest in the gaps between panel rows (Table 3).

Specifically, at a 5 cm depth, the average Ts at the RF site is approximately 2.33°C higher than beneath the panels (p<0.001, paired t-test) and 3.83°C higher than in the gaps (p<0.001, paired t-test). At 10 cm depth, the temperature difference is approximately 1.89°C higher at the RF site compared to beneath the panels (p<0.001, paired t-test), and 3.65°C higher compared to the gaps (p<0.001, paired t-test). The slightly higher Ts beneath the PV panels compared to the gaps may be attributed to the insulation effect of the panels and their thermal radiation properties. PV panels reduce direct exposure to cold air, limiting heat loss from the soil. Additionally, the panels absorb solar radiation and transfer some of the heat to the soil beneath, helping to maintain relatively higher temperatures. In contrast, the soil in the gaps between the rows is more exposed to cold air, leading to greater temperature fluctuations and lower overall temperatures. (Yue et al., 2021) also reported that the soil beneath the PV panels maintains warmer soil conditions in winter due to the insulation and heat transfer provided by the PV panels.

Soil volumetric water content (VWC) also exhibits significant seasonal variations across different locations (Figure 8, Table 4). In spring and autumn, soil moisture at the RF site remains consistently higher compared to the other two locations (p<0.001, paired t-test) (Table 4). This is primarily due to earlier soil thawing in spring and delayed freezing in autumn at the RF site. In summer, average SWC in gaps between the PV rows is higher than that at the RF site by about 9% at both depths (p<0.001, paired t-test) (Table 4). This difference can be mainly attributed to two main factors: (1) the inclined structure of the PV panels channels precipitation toward the gaps between the rows, significantly enhancing water recharge in this area; (2) the lower wind speed and reduced soil temperature within gaps between the rows, caused by shading from the PV power plant, effectively suppress evaporation and maintain higher soil moisture levels. This result align with Choi et al. (2024), who reported that the interspace of PV power plant had the highest soil moisture (25 cm) regardless of whether the PV was bare or vegetated across three utility-scale PV facilities in Minnesota, USA.

**Table 4.** Average soil moisture at 5 cm and 10 cm depths for PV and RF sites.

|  | Annual | Spring | Summer | Fall | Winter |
|---|---|---|---|---|---|
| RF site | 0.36 | 0.38 | 0.44 | 0.46 | 0.15 |
| (5cm) | (±0.001) | (±0.001) | (±0.001) | (±0.001) | (±0.001) |
| PV site | 0.37 | 0.41 | 0.48 | 0.45 | 0.12 |
| (gap, 5cm) | (±0.001) | (±0.002) | (±0.001) | (±0.002) | (±0.001) |
| PV site | 0.27 | 0.29 | 0.37 | 0.32 | 0.10 |
| (beneath, 5cm) | (±0.001) | (±0.001) | (±0.001) | (±0.001) | (±0.001) |
| RF site | 0.40 | 0.44 | 0.47 | 0.51 | 0.18 |
| (10cm) | (±0.001) | (±0.001) | (±0.001) | (±0.001) | (±0.001) |
| PV site | 0.37 | 0.36 | 0.51 | 0.48 | 0.12 |
| (gap, 10cm) | (±0.001) | (±0.002) | (±0.001) | (±0.002) | (±0.001) |
| PV site | 0.35 | 0.38 | 0.47 | 0.43 | 0.10 |
| (beneath, 10cm) | (±0.001) | (±0.002) | (±0.001) | (±0.001) | (±0.001) |

Conversely, the summer average SWC beneath the PV panels is the lowest among the three locations (Table 4) due to precipitation interception by the panels, which limits direct water input to the underlying soil. Interestingly, at 10 cm depth, the summer average SWC shows no significant difference between the beneath-PV and RF sites ($p>0.05$). However, at 5 cm depth, the SWC beneath PV panels is approximately 16% lower than at the RF site ($p<0.001$, paired t-test). This suggests that subsurface soil layers under PV panels may benefit from lateral water redistribution and upward soil moisture migration driven by capillary action and vegetation root water uptake (Jury and Horton, 2004), partially offsetting the reduced surface recharge. Different form the result of Yue et al. (2021), who reported significantly higher SWC at depths of 10–40 cm beneath PV panels compared to non-PV areas during the rainy season, with the moisture difference decreasing with depth. Choi et al. (2024) observed inconsistent effects across facilities: the median SWC beneath PV panels was lower than the reference site in two facilities but higher in one. This highlights that, even under similar climatic conditions, variations in PV system structure, soil properties, and management practices can lead to inconsistent SWC patterns beneath PV panels.

Due to the seasonal variability of precipitation, seven consecutive 6-day periods without rainfall occurred in this alpine meadow when the soil was in a thawed state. The linear decline slopes of daily SWC at different locations inside and outside the PV power plant were compared. The Figure 9 shows that the decrease rates of soil moisture

at the RF site at 5 cm and 10 cm depths are significantly higher than those at PV sites, approximately 1.3 times and 3.5 times the rate of decrease between the PV rows (p<0.05, paired t-test) and beneath PV panels (p<0.001, paired t-test), respectively. These results further indicate that PV power plant can effectively mitigate topsoil moisture loss during dry periods.

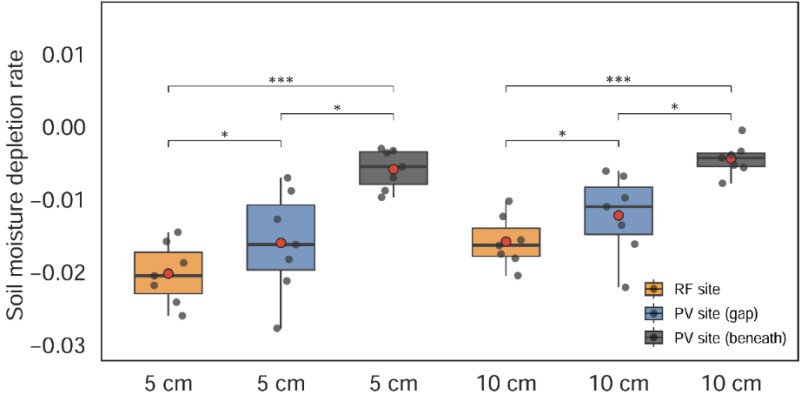

**Figure 9.** Soil moisture depletion rates at 5 cm and 10 cm depths for the RF site and PV site (gap and beneath panels) during dry periods. Asterisks indicate statistically significant differences (*p < 0.05, **p < 0.01, ***p < 0.001).

To further investigate the mechanisms underlying these spatial differences, we applied a Boosted Regression Tree (BRT) model to quantify the relative importance of microclimatic variables for each site (Figure 10). At the RF site, near-surface VPD at 2.5 m and 10 m were the dominant predictors, jointly accounting more than 60% of the total importance (40.1% and 16.0%, respectively), reflecting the strong influence of atmospheric evaporative demand in open grassland. In contrast, the PV gap exhibited a shift toward energy-limited regulation, with shallow soil temperature at 5 cm (42.9%) and air temperature at 2.5 m (23.6%) identified as the primary controls, reflecting the effects of modified radiation regimes on surface energy balance. Beneath the PV panels, VPD at 2.5 m again emerged as the most important driver (40.6%), suggesting a reestablishment of atmospheric control on soil moisture dynamics despite persistent shading.

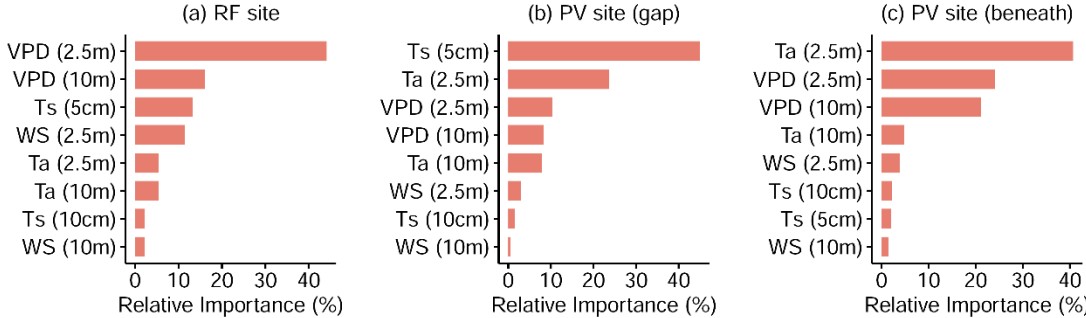

**Figure 10.** Relative importance of predictor variables in explaining changes in daily soil moisture during dry periods at (a) the RF site, (b) the PV site (gap), and (c) the PV site (beneath panels), based on boosted regression tree (BRT) analysis. The predictor variables include vapor pressure deficit (VPD) at 2.5 m and 10 m, air temperature (Ta) at 2.5 m and 10 m, soil temperature (Ts) at 5 cm and 10 cm, and wind speed (WS) at 2.5 m and 10 m.

These spatial differences in variable importance point to fundamental shifts in the microclimatic drivers of soil moisture across the landscape. At the RF site, unshaded conditions allow for high solar input and strong turbulent mixing, promoting tight coupling between the surface and atmosphere. This coupling enhances air temperature and vapor pressure deficit (VPD), thereby driving strong evaporative demand and rapid soil drying during dry spells. In the PV gap, intermittent shading modifies the radiation regime by reducing net radiation reaching the soil surface. As a result, soil and air temperature become more effective indicators of the constrained energy supply and emerge as dominant controls of soil moisture variability. This transition aligns with the observed soil moisture depletion rates (Fig. 9), where the PV gap exhibited significantly slower drying than the RF site, suggesting a partial decoupling from atmospheric demand. Beneath the panels, the mechanisms are more complex. Although direct shortwave radiation is strongly reduced, the thermal environment is influenced by convective and longwave radiation emitted from the heated PV panel surfaces. This process elevates near-surface air temperature, which in turn sustains relatively high VPD. As a result, VPD remains the dominant driver of soil moisture dynamics, not solely through ambient climate forcing but also via a structurally mediated atmospheric control pathway. Thus, beneath-panel soil moisture loss remains demand-limited, but the evaporative demand itself is shaped by infrastructure-induced microclimatic feedbacks.

## 4. Conclusions and Implications

Pastoral-integrated PV power plant, a form of agrivoltaism, offer an innovative solution to the increasing demand for sustainable pastoralism and renewable energy. As solar PV energy development continues to expand, concerns about its potential impacts on the ecological environment, particularly in fragile and sensitive regions such as the Tibetan Plateau, are gaining prominence.

This study investigates the effects of PV power plant on the local meteorological conditions and soil hydrothermal dynamics in a high-altitude alpine meadow on the eastern Tibetan Plateau. The findings reveal that the installation of PV panels increases annual mean net radiation by 28.9%, while reducing albedo and 2.5 m wind speed by 31.6% and 36.2%, respectively. Despite the slight warming effect on annual mean air temperature (Ta), the impact of PV power plant is highly asymmetric: daytime heating, nighttime cooling, summer warming, and winter cooling. During summer daytime, Ta, q, and VPD at the PV site are 11.1%, 4%, and 22.2% higher, respectively, than at the RF site. The PV power plant introduces substantial spatial heterogeneity in soil hydrothermal properties. At a depth of 5 cm, compared to the RF site, the annual soil temperature (Ts) decreased by approximately 45.7% in the PV array gaps and 54.3% beneath the panels, while the annual soil water content (SWC) increased by about 2.8% in the gaps but decreased by 27.1% beneath the panels. Lower soil temperatures extend the frozen period within the PV power plant by nearly 50 days compared to the RF site. Furthermore, the rate of soil moisture loss during non-freezing periods is significantly reduced (p<0.001), with depletion rates beneath the panels and in the gaps being up to 3.5 and 1.3 times lower, respectively, compared to the RF site. Taken together, these microclimatic and soil hydrothermal changes induced by the PV power plant pave the way for exploring their broader ecological implications.

Climate projections indicate that global temperatures will likely surpass the 1.5 °C threshold, with regional warming on the Tibetan Plateau expected to exceed the global average due to elevation-dependent warming (You et al., 2020). Such rapid warming has already impacted soil freeze-thaw cycles, hydrological systems, carbon dynamics,

and vegetation succession (Armstrong et al., 2014; Chen et al., 2022; Luo et al., 2020; Ma et al., 2022; Yang et al., 2014; Zhao et al., 2021). By buffering soil temperature fluctuations and extending the frozen period, PV power plant may help delay soil thawing and potentially reduce permafrost degradation risks, although direct measurements of permafrost dynamics are needed to confirm this possibility. Moreover, reduced soil moisture depletion during dry periods potentially creates more favorable conditions for vegetation growth and photosynthesis. However, the shortened growing season may reduce vegetation carbon absorption, potentially offsetting some of these benefits. Thus, to verify these potential impacts on ecosystem carbon exchange, plant physiological responses and greenhouse gas fluxes need to be systematically measured.

The study area, a critical water source region for the upper Yellow River, plays a vital role in regional hydrology. The observed increase in soil moisture and slower depletion during dry periods may promote deeper percolation, potentially enhancing groundwater recharge and supporting water conservation. However, changes in freeze-thaw dynamics, such as advanced freezing and delayed thawing, may alter the spatial and temporal distribution of runoff by affecting the release of meltwater and frozen soil water, with potential consequences for downstream water availability. Although the current observational equipment in this study does not provide quantitative evapotranspiration (ET) results, an analysis based on the established relationship between ET and meteorological factors (Allen et al., 1998) indicates that PV power plant may exert a dual impact on ET. On one hand, PV power plant elevate Ta and VPD, which increase atmospheric demand for water vapor and may potentially intensify ET. On the other hand, the shading effect of PV power plant reduces the Rn reaching the soil surface, and the structural design of the arrays significantly decreases wind speed, both of which tend to suppress ET. Therefore, subsequent research should utilize eddy covariance systems, along with lysimeters, to quantitatively assess the impact of PV power plant on ET.

From a biodiversity perspective, the introduction of microenvironmental heterogeneity through shading and moisture redistribution by PV power plant may influence vegetation dynamics. Observed reductions in shallow soil temperature and prolonged

soil frozen periods in this PV power plant may reduce the competitive advantage of tall, light-demanding grasses, potentially creating niches for shade-tolerant C3 plants (Wang et al., 2024a). Such shifts might counteract vegetation homogenization trends. However, the shortened unfrozen period may pose challenges for species with growth cycles tied to freeze-thaw dynamics, potentially reducing their adaptive capacity. Overall, the PV

power plant may exert both stabilizing and destabilizing influences on plant communities. To verify these implications, future studies should incorporate species richness, plant growth, and plant trait composition into long-term monitoring frameworks.

These broader-scale implications suggest that PV power plants may offer a pathway to

600 enhance the resilience of alpine meadow ecosystems in a warming climate. However, given the ecological fragility of the Tibetan Plateau, the complexity of ecosystem feedbacks to both climate change and human activities, and the limited duration and scope of observations in this study, the sustainable and ecologically compatible development of PV infrastructure in this region still requires comprehensive

investigation. Future research should incorporate multi-year observations and numerical modeling to assess the long-term effects of PV power plants on regional climate, hydrological processes, and ecological functions, including biodiversity and carbon sequestration, under varying climatic conditions.

**Code/Data availability**

The data and code supporting the findings of this study are available upon request to the corresponding author, Shaoying Wang (wangshaoying@lzb.ac.cn).

**Author contribution**

Conceptualization: SW and XM; methodology: SW, PY and QL; funding acquisition: SW, XM, ZL and ZL; data curation and visualization: SW, LS, and WN; field

measurement: SW, WN; All authors actively contributed to the discussions and to the writing of the final version of the paper.

**Competing interests**

The contact author has declared that none of the authors has any competing interests.

**Acknowledgments**

The authors would like to thank the Zoige Plateau Wetland Ecosystem Research Station for providing the observation equipment used in this study. We are also deeply grateful to the editor and reviewers for their constructive comments, which greatly improved the quality of this paper.

**Financial support**

This study was supported by the National Science Fund for Distinguished Young Scholars of China (42325502), the Key Project of Gansu Province Science and Technology Plan (23ZDFA011), the National Natural Science Foundation of China (41875018), the Science and Technology Research Plan of Gansu Province (22JR5RA048), and the Ningxia Hui Autonomous Region Key Research and Development Program (2024BEG03003).

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
