# Peer review of "Environmental Impacts of Pastoral-Integrated Photovoltaic Power Plant in an Alpine Meadow on the Eastern Tibetan Plateau"

_EGUsphere, 2025_

## Author Response (AR1)

**Response to reviewers' comments:**

We would like to thank reviewers for their thoughtful and constructive comments on our manuscript over the two rounds of review. We greatly appreciate the time and effort they have put into reviewing our work. Their feedback has been invaluable in improving the quality and clarity of our research. Below, we provide a detailed response to Reviewers' comments and outline the revisions we have made to address their concerns.

**Reviewer #1:**

This manuscript investigates the environmental impacts of a pastoral-integrated photovoltaic (PV) solar farm in an alpine meadow ecosystem on the eastern Tibetan Plateau. The authors conduct a one-year field study comparing microclimate and soil hydrothermal conditions beneath a PV array and in an adjacent undisturbed reference meadow. At both sites, the study examines changes in net radiation, albedo, air and soil temperature, wind speed, soil moisture, and freeze-thaw dynamics. The PV array significantly alters these variables, and it is found that net radiation is higher, daytime temperatures increase while nighttime temperatures decrease, wind speed is reduced, and soil freeze-thaw timing and soil moisture are modified. The authors interpret these shifts as potential benefits for ecosystem resilience, suggesting that PV arrays may buffer against climate warming, enhance soil water availability, and support vegetation growth.

The study makes a timely contribution to the emerging field of agrivoltaics and provides valuable empirical data from a high-altitude grassland ecosystem. The manuscript is generally well written. The topic is both interesting and of scientific and societal importance, making it potentially suitable for publication in ACP. But the methods employed in the study lack sufficient robustness. Several other critical aspects need improvement before the manuscript can be considered for publication. Additionally, the discussion/implication is largely hypothetical and weakly supported by the presented results. I detail my specific comments below.

**Response:** We thank Reviewer 1 for recognizing the timely contribution of our study to the field of agrivoltaics and for highlighting its scientific and societal importance.

We have carefully addressed the two overarching concerns by enhancing methodological robustness through detailed inferential analyses, uncertainty estimates and clarifications of key methods such as the definition of net radiation and the handling of missing data. We have also ensured that ecological implications are clearly labeled as hypotheses where appropriate. Below, we provide a point-by-point response to each of your comments.

**Comments 1**. The authors analyzed seasonal and annual averages of measured microclimatic variables at both sites, and reported relative differences. They also derived specific metrics such as the number of days when soil is frozen or thawed, and linear trends of daily soil moisture decline. Some useful comparisons are made with previous studies. While these calculated outcomes are useful for interpreting microclimate impacts, the results remain largely descriptive. More in-depth quantitative and inferential analyses are expected, particularly regarding the relationship between these variables, and/or how the climate-vegetation, soil-plant feedbacks are influenced. Moreover, some of the analytical methods lack sufficient details or need clarification. For example, the definition of net radiation Rn, and how missing data were handled when computing averages.

**Response to Comments 1:** We thank the reviewer for this thoughtful and constructive comment. In response, we have taken several steps to clarify our methodology and deepen our analysis:
(1) Clarification of analytical methods
We have added detailed descriptions of the data gap-filling procedures in the revised manuscript (Section 2.3, Lines 189–194). Specifically: "As a result of these procedures, over 99% of the 10-minute microclimatic and soil hydrothermal data from June 2023 to May 2024 were retained, with no continuous gaps exceeding 1 hours. Missing values were filled via linear interpolation using a centered 12-point (2-hour) moving window. Daily means were calculated only when at least 75% of the 10-minute data were available. Seasonal and annual means were then computed from these daily averages

without further interpolation."

We also clarified the definition of net radiation (Rn = DR + DLR – UR – ULR) and surface albedo (α = UR / DR) in the revised manuscript (Section 2.4, Lines 195–202), where DR, DLR, UR, and ULR represent downward shortwave radiation, downward longwave radiation, upward shortwave radiation, and upward longwave radiation, respectively (Section 2.3, Lines 196–202).

(2) Enhancement of quantitative and inferential analysis:

To move beyond descriptive statistics, we introduced a Boosted Regression Tree (BRT) model to quantify the relative importance of key microclimatic variables (e.g., air temperature, vapor pressure deficit, soil temperature, and wind speed) in explaining soil moisture variability during dry periods at different microsites. A detailed description of BRT method has been added in the revised manuscript (Section 2.4, Lines 205–215).

As shown in Section 3.4 (Lines 493–530 and Figure 10). The main results of BRT model revealed distinct control regimes: At the RF site, VPD at 2.5m and 10m dominated the model (40.1% and 16.0%), indicating strong atmospheric demand–limited control; In the PV gap, shallow soil temperature (42.9%) and near-surface air temperature (23.6%) were the main predictors, suggesting an energy-limited regime driven by surface heating; Beneath the PV panels, VPD (40.6%) and Ta (24.0%) at 2.5m were dominant, reflecting a structurally buffered but still demand-limited environment.

**Comments 2.** No uncertainty estimates (e.g., standard deviations or confidence intervals) are provided alongside the reported means or differences. Moreover, the use of the term "significantly" (e.g., Lines 266 and 337) implies statistical significance, yet no statistical tests (such as t-tests) or significance thresholds are specified. This issue should be addressed consistently throughout the manuscript to ensure clarity and scientific rigor.

**Response to Comments 2:** We thank the reviewer for this important comment. In the revised manuscript, we have systematically addressed this issue to ensure scientific rigor. Specifically, we have:

Added uncertainty estimates (standard deviations) alongside all reported mean values

throughout the Results section and in all relevant tables and figures (e.g., Tables 1–4; Figures 5–7).

Conducted paired Student's t-tests to assess the statistical significance of differences between the PV and RF sites for key variables, including air temperature, specific humidity, vapor pressure deficit, wind speed, radiation components, soil temperature, and soil moisture (e.g., Tables 1; Figures 3,5,6,7,9). Revised all instances where the term "significantly" was previously used without statistical support. The term is now only applied when supported by statistical tests.

Clearly specified in the Methods section (Section 2.4, Lines 203–205) the statistical tests applied.

**Comments 3**. The most critical issue is that the design constraints limit the generality of the findings. Only a single PV plot and one control plot are monitored, so all analyses are based on this one "case study" comparison. It may not robustly distinguish PV effects from site-specific variation. Without multiple PV–reference pairs or randomized treatment plots, the study can be biased by any observed differences due to unmeasured local peculiarities rather than the PV installation. For example, small differences in soil type, compaction, or micro-topography between the sites could influence soil moisture and temperature independently of the panels. The authors do not report characterizing baseline similarities (e.g. detailed vegetation coverage and composition, or soil texture) between the PV and reference sites, making it hard to rule out confounding factors.

**Response to Comments 3:** We appreciate the reviewer's concern regarding potential confounding effects due to the spatial separation between the PV and RF sites. To address this, we have included additional information in the revised manuscript (Section 2.1, Lines 136–164) to clarify the comparability between the two sites. Both towers were located within the same contiguous open alpine meadow, with no fencing or physical barriers, and were subjected to identical grazing regimes under free-ranging sheep grazing from May to September. The average vegetation height (0.3 m in summer, 0.1 m in winter) was consistent across the landscape. Soil samples from both sites were classified as sandy loam, and statistical tests indicated no significant difference in bulk

density at 5 cm depth among the locations. At 10 cm depth, slightly higher bulk density was observed between panel rows, which may be attributed to construction-related compaction, but values at the RF site and beneath panels were statistically comparable. Furthermore, vegetation surveys during peak growing season revealed high coverage (>95%) and a similar dominant species composition across all sites.

Taken together, these observations suggest that the PV and RF sites shared similar baseline soil and vegetation characteristics, and that the observed microclimatic and soil hydrothermal differences are unlikely to be artifacts of pre-existing spatial variability. Rather, they reflect the structural and radiative modifications introduced by the PV installation.

**Comments 4**. Although the term "pastoral-integrated" in the title implies ongoing grazing, the paper does not describe grazing management at the sites. Differences in grazing pressure or trampling under versus outside the arrays could alter soil structure and vegetation, yet this is not addressed.

**Response to Comments 4:** We thank the reviewer for this insightful comment. In the revised manuscript (Section 2.1, Lines 141–145), we have added a detailed description of the grazing management to clarify that both the PV and RF sites were subjected to the same grazing pressure. Specifically, we note that free-ranging sheep grazing by local herders occurred uniformly across the entire study area, including both within and outside the PV arrays, from May to September each year. There are no physical barriers, fences, or grazing management differences between the two sites. Given that both sites were subject to identical grazing regimes and stocking densities, we believe that differences in trampling or grazing intensity are unlikely to have introduced significant bias in soil structure or vegetation composition between the PV and reference plots.

**Comments 5**. Many of the ecosystem implications claimed (resilience to warming, permafrost protection, enhanced biodiversity) are unsubstantiated by the presented data or existing studies. Care should be taken to clearly distinguish measured microclimate/soil changes from conjecture about ecological impacts. Linking abiotic

changes to biotic responses requires additional studies (e.g. measuring plant growth, soil carbon, hydrology under the arrays). Some claims or ecological interpretations (e.g. "mitigate permafrost degradation", "improve alpine meadow ecosystem resilience") may far overreach the measured data. More cautious phrasing or acknowledgment of uncertainties would be necessary.

**Response to Comments 5:** We thank the reviewer for this important and constructive comment. In the revised manuscript, we have carefully revised all statements concerning ecosystem implications by using more cautious phrasing and emphasizing the need for future field experiments. These revisions ensure that the discussion is appropriately cautious and clearly distinguishes between measured microclimatic or soil hydrothermal changes and their potential ecological consequences. Specifically:

The sentences "By buffering soil temperature fluctuations and extending the frozen period, PV arrays can mitigate permafrost degradation and associated greenhouse gas emissions. Moreover, reduced soil moisture depletion during dry periods creates favorable conditions for vegetation growth and photosynthesis. However, the shortened growing season may reduce vegetation carbon absorption, potentially offsetting some of these benefits." have been revised to: "By buffering soil temperature fluctuations and extending the frozen period, PV power plant may help delay soil thawing and potentially reduce permafrost degradation risks, although direct measurements of permafrost dynamics are needed to confirm this possibility. Moreover, reduced soil moisture depletion during dry periods potentially creates more favorable conditions for vegetation growth and photosynthesis. However, the shortened growing season may reduce vegetation carbon absorption, potentially offsetting some of these benefits. Thus, to verify these potential impacts on ecosystem carbon exchange, plant physiological responses and greenhouse gas fluxes need to be systematically measured." (Section 4, Lines 561–569).

The sentences "Enhanced soil moisture promotes deep percolation, which strengthens groundwater recharge and supports water conservation. However, changes in freeze-thaw dynamics, such as advanced freezing and delayed thawing, could alter the spatial and temporal distribution of runoff. This is particularly evident in the release patterns

of meltwater and frozen soil water, potentially disrupting downstream water availability." have been revised to:"The observed increase in soil moisture and slower depletion during dry periods may promote deeper percolation, potentially enhancing groundwater recharge and supporting water conservation. However, changes in freeze-thaw dynamics, such as advanced freezing and delayed thawing, may alter the spatial and temporal distribution of runoff by affecting the release of meltwater and frozen soil water, with potential consequences for downstream water availability." (Section 4, Lines 571–576).

The sentences "The reduced Ts and longer frozen conditions may limit the dominance of tall, light-demanding grasses, creating niches for shade-tolerant C3 plants (Wang et al., 2024a). This shift could counteract vegetation homogenization trends, promoting species diversity in alpine grasslands. However, the shortened growing season poses challenges for species with growth cycles tied to freeze-thaw dynamics, potentially reducing their adaptive capacity. PV arrays may act as both stabilizing and destabilizing forces for plant communities, necessitating further study on their long-term ecological impacts." have been revised to: "Observed reductions in shallow soil temperature and prolonged soil frozen periods in this PV power plant may reduce the competitive advantage of tall, light-demanding grasses, potentially creating niches for shade-tolerant C3 plants (Wang et al., 2024a). Such shifts might counteract vegetation homogenization trends. However, the shortened unfrozen period may pose challenges for species with growth cycles tied to freeze-thaw dynamics, potentially reducing their adaptive capacity. Overall, the PV power plant may exert both stabilizing and destabilizing influences on plant communities. To verify these implications, future studies should incorporate species richness, plant growth, and plant trait composition into long-term monitoring frameworks." (Section 4, Lines 589–598).

**Editorial comments:**

i) All figure captions should be improved and expanded to clearly describe the information presented in the figures.

ii) Rephrase the sentence in Line 357 for clarity.

iii) Modify the figure and table numbering format throughout the manuscript: change "Figs. 01, 02, 03, …" and "Tables 01, 02, …" to "Figs. 1, 2, 3, …" and "Tables 1, 2, …" respectively, in accordance with standard formatting conventions. Labels are missing in Fig. 1.

**Response to Editorial comments:** We thank the reviewer for these helpful editorial suggestions. All the recommended revisions have been carefully addressed in the revised manuscript:

Figure captions have been improved and expanded to clearly describe the variables presented, measurement units, and key comparisons depicted in each figure.

The sentence on Line 357 in original manuscript has been rephrased to: "Wintertime Ts shows a distinct spatial pattern, with values being highest at the RF site, intermediate beneath the panels, and lowest in the gaps between panel rows (Table 3)." (Section 3.4, Lines 433–435).

The figure and table numbering formats have been corrected throughout the manuscript to comply with Atmospheric Chemistry and Physics journal style guidelines, now using "Figures. 1, 2, 3…" and "Tables 1, 2…" instead of the previously used "Figs. 01, 02…" format. In addition, we have added the missing labels to Fig. 1 to improve readability and orientation.

**Reviewer #2:**

This manuscript presents a well-structured and insightful investigation into the environmental impacts of a pastoral-integrated photovoltaic (PV) power plant in the ecologically sensitive alpine meadows of the eastern Tibetan Plateau. The authors combine detailed in-situ observations over a full annual cycle to assess microclimatic and soil hydrothermal changes induced by the PV installation. The study is timely, relevant, and contributes significantly to the emerging literature on the ecological footprint of renewable energy infrastructure in high-altitude regions.

However, while the paper is methodologically sound and generally well-written, there are several issues that need to be addressed to improve clarity, scientific rigor, and impact. Specific suggestions are listed below:

**Comments 1**. The ecological conclusions regarding biodiversity and vegetation dynamics are speculative, as no direct measurements of plant composition, cover, or productivity were made. The authors should indicate which conclusions are hypothesized rather than observed, and consider including at least some basic vegetation survey data if available.

**Response to Comments 1:** We appreciate the reviewer's thoughtful comment. In the revised manuscript, we have taken several steps to address this concern:

(1) We have included results from a basic vegetation survey conducted during the peak growing season using 0.3 m × 0.3 m quadrats at each microsite. The results showed that vegetation cover exceeded 95% at all positions (RF site, beneath PV panels, and between panel rows), and the dominant species were consistent across microsites, including Stipa aliena, Potentilla anserina, and Scirpus pumilus. These details have been added to Section 2.1 (Lines 162–164).

(2) To ensure a clear distinction between measured results and their potential ecological consequences, we have carefully revised all statements concerning ecosystem implications by adopting more cautious phrasing and emphasizing the need for future field experiments. Specifically:

The sentences "The reduced Ts and longer frozen conditions may limit the dominance of tall, light-demanding grasses, creating niches for shade-tolerant C3 plants (Wang et al., 2024a). This shift could counteract vegetation homogenization trends, promoting species diversity in alpine grasslands. However, the shortened growing season poses challenges for species with growth cycles tied to freeze-thaw dynamics, potentially reducing their adaptive capacity. PV arrays may act as both stabilizing and destabilizing forces for plant communities, necessitating further study on their long-term ecological impacts." have been revised to "Observed reductions in shallow soil temperature and prolonged soil frozen periods in PV power plant may reduce the competitive advantage of tall, light-demanding grasses, potentially creating niches for shade-tolerant C3 plants (Wang et al., 2024a). Such shifts might counteract vegetation homogenization trends. However, the shortened unfrozen period may pose challenges for species with growth cycles tied to freeze-thaw dynamics, potentially reducing their adaptive capacity.

Overall, the PV power plant may exert both stabilizing and destabilizing influences on plant communities. To verify these ecological implications, future studies should incorporate species richness, plant growth, and plant trait composition into long-term monitoring frameworks." (Section 4, Lines 589–598).

**Comments 2**. While the discussion of hydrology and carbon cycling is extensive, the study does not provide direct measurements of evapotranspiration (ET), carbon fluxes, or runoff.

**Response to Comments 2:** We thank the reviewer for this important point. We fully acknowledge that this study did not include direct measurements of evapotranspiration (ET), carbon fluxes (e.g., GPP, NEE), or surface runoff. In the revised manuscript, we have taken the following steps to address this concern:

(1) We have explicitly stated that the effects of PV arrays on ET were inferred from key meteorological drivers (i.e., Rn, Ta, VPD, and wind speed) based on established relationships described in the Penman-Monteith framework (Allen et al., 1998), rather than directly measured. We now clearly indicate that the dual effects of PV arrays on ET (i.e., enhanced atmospheric demand vs. reduced available energy and wind speed) are conceptual interpretations. These interpretations require further validation through direct observations using eddy covariance systems and lysimeters (Section 4, Lines 576–586).

(2) We have clarified that no carbon fluxes measurements (e.g., GPP, NEE) were conducted in this study. While the observed changes in microclimate and soil hydrothermal dynamics may have implications for carbon and water cycling, these potential effects require dedicated flux measurements in future research for verification.

(3) To avoid overinterpretation, we have revised the language to make these hydrological and ecological implications conditional and hypothesis-generating, rather than conclusive. Specifically:

The sentences "Moreover, reduced soil moisture depletion during dry periods creates favorable conditions for vegetation growth and photosynthesis. However, the shortened growing season may reduce vegetation carbon absorption, potentially offsetting some

of these benefits" have been revised to "Moreover, reduced soil moisture depletion during dry periods potentially creates more favorable conditions for vegetation growth and photosynthesis. However, the shortened growing season may reduce vegetation carbon absorption, potentially offsetting some of these benefits. Thus, to verify these potential impacts on ecosystem carbon exchange, plant physiological responses and greenhouse gas fluxes need to be systematically measured." (Section 4, Lines 564–569). The sentences "Enhanced soil moisture promotes deep percolation, which strengthens groundwater recharge and supports water conservation. However, changes in freeze-thaw dynamics, such as advanced freezing and delayed thawing, could alter the spatial and temporal distribution of runoff. This is particularly evident in the release patterns of meltwater and frozen soil water, potentially disrupting downstream water availability." have been revised to "The observed increase in soil moisture and slower depletion during dry periods may promote deeper percolation, potentially enhancing groundwater recharge and supporting water conservation. However, changes in freeze-thaw dynamics, such as advanced freezing and delayed thawing, may alter the spatial and temporal distribution of runoff by affecting the release of meltwater and frozen soil water, with potential consequences for downstream water availability." (Section 4, Lines 571–576).

**Comments 3.** The paper attributes observed microclimatic and soil changes to PV panels, but the distance between the PV and RF sites (180 m) might allow confounding from spatial variability.

**Response to Comments 3:** We thank the reviewer for raising this important concern. In the revised manuscript (Section 2.1), we have expanded our description of the study design to clarify the spatial relationship and environmental comparability between the PV and reference (RF) sites. Specifically:

(1) Site continuity and distance: The PV and RF towers are within the same contiguous alpine meadow. There are no fences, landform breaks, or vegetation boundaries between the sites, and both are situated on flat terrain with similar slope and aspect (Section 2.1, Lines 138–140).

(2) Uniform land use and grazing regime: Both sites are subject to identical land use practices, including free-ranging sheep grazing by local herders between May and September. There are no site-specific differences in stocking density or grazing management. Seasonal vegetation height was also similar across sites (0.3 m in summer, 0.1 m in winter) (Section 2.1, Lines 140–145).

(3) Baseline comparability of soil and vegetation: Soil samples collected at 5 cm and 10 cm depths revealed both PV and RF sites were classified as sandy loam. Bulk density at 5 cm showed no significant difference ($p > 0.05$), while a slight increase was observed at 10 cm between panel rows, likely due to construction compaction rather than background heterogeneity. Additionally, a vegetation survey during the peak growing season (based on $0.3 \times 0.3$ m quadrats) showed >95% ground cover and consistent dominant species (Stipa aliena, Potentilla anserina, and Scirpus pumilus) across microsites (Section 2.1, Lines 152–164).

**Comments 4.** While the study touches on ecosystem resilience and hydrological contributions, the policy relevance is underdeveloped.

**Response to Comments 4:** We thank the reviewer for this helpful comment. We agree that highlighting the policy relevance of our findings is important, especially in the context of balancing renewable energy development with ecological protection in sensitive regions such as the Tibetan Plateau. While our study touches on the potential implications of PV power plant development for ecosystem resilience and hydrological processes, given the ecological fragility of the Tibetan Plateau, the complexity of ecosystem feedbacks to both climate change and human activities, and the limited duration and scope of observations in this study, we consider it would be premature to draw strong policy conclusions at this stage. In the revised manuscript, we have revised the following statement to reflect the broader implications of our findings and acknowledge their potential relevance to future land-use and renewable energy planning:

"These broader-scale implications suggest that PV power plants may offer a pathway to enhance the resilience of alpine meadow ecosystems in a warming climate. However,

given the ecological fragility of the Tibetan Plateau, the complexity of ecosystem feedbacks to both climate change and human activities, and the limited duration and scope of observations in this study, the sustainable and ecologically compatible development of PV infrastructure in this region still requires comprehensive investigation. Future research should incorporate multi-year observations and numerical modeling to assess the long-term effects of PV power plants on regional climate, hydrological processes, and ecological functions, including biodiversity and carbon sequestration, under varying climatic conditions." (Section 2.1, Lines 599–608).

**Comments 5.** The abstract is rich in content but slightly too long. In addition, some data from the conclusion should complement these parts.

**Response to Comments 5:** We thank the reviewer for this helpful suggestion. In the revised manuscript, we have carefully shortened and refined the abstract to enhance clarity and focus. The revised abstract is as follows:

Abstract: Rising global energy demand and the transition to low-carbon sources have driven the rapid expansion of photovoltaic (PV) power plants, introducing significant land-use changes with largely unexplored ecological consequences. This study examined the microclimatic and soil hydrothermal impacts of a pastoral-integrated PV power plant in an alpine meadow ecosystem on the eastern Tibetan Plateau. Year-round observations from two 10-meter towers, located inside and outside the PV power plant, revealed that PV installations increased annual net radiation by 28.9%, while reducing albedo and wind speed by 31.6% and 36.2%, respectively. Air temperature responses were highly asymmetrical, with daytime and summer warming but nighttime and winter cooling. The PV power plant induced strong spatial heterogeneity in soil temperature and moisture: At 5cm depth, PV array gaps exhibited cold and moist conditions, whereas areas beneath the panels were cold and dry. These changes extended the soil frozen period by approximately 50 days and reduced the soil moisture depletion rate by 1.3 to 3.5 times compared to the reference site. These findings indicate that the PV power plant can alter local energy and water balances in ways that may buffer ecosystem responses to climate warming. However, further multi-year studies are

needed to evaluate their long-term impacts on vegetation dynamics, carbon fluxes, and permafrost processes.

**Comments 6.** "cold-dry" and "cold-moist" could be better defined quantitatively (e.g., relative to the reference site).

**Response to Comments 6:** We thank the reviewer for this valuable suggestion. In the revised manuscript, we have adopted a more quantitative description to clarify the soil hydrothermal changes induced by the PV power plant. Specifically, the original sentence "Beneath the panels, the soil exhibits a cold-moist characteristic, while the gaps between PV rows display a cold-dry distribution." has been revised to "At a depth of 5 cm, compared to the RF site, the annual soil temperature (Ts) decreased by approximately 45.7% in the PV array gaps and 54.3% beneath the panels, while the annual soil water content (SWC) increased by about 2.8% in the gaps but decreased by 27.1% beneath the panels." (Section 4, Lines 546–549).

**Comments 7.** In line 157, " Data were collected continuously over a one-year period, from June 2023 to May 2023…"— likely a typo.

**Response to Comments 7:** We thank the reviewer for pointing out this typographical error. The correct time period is "from June 2023 to May 2024," and this has been corrected in the revised manuscript (line 180).

**Comments 8.** In the abstract, inconsistent terminology such as "ground-mounted solar parks," "PV power station," and "PV plant" is used. These terms should be unified throughout the article. If "PV plant" is preferred, it should be corrected to the more accurate term "PV power plant."

**Response to Comments 8:** We appreciate the reviewer's suggestion regarding the inconsistent terminology. In the revised manuscript, we have standardized the terminology throughout the text. We now consistently use the term "PV power plant", which we agree is the most accurate and appropriate in this context.